# Fighting Boredom in Recommender Systems with Linear Reinforcement Learning

**Romain Warlop**
fifty-five, Paris, France
SequeL Team, Inria Lille, France
romain@fifty-five.com

**Alessandro Lazaric**
Facebook AI Research
Paris, France
lazaric@fb.com

**Jérémie Mary**
Criteo AI Lab
Paris, France
j.mary@criteo.com

## Abstract

A common assumption in recommender systems (RS) is the existence of a *best fixed* recommendation strategy. Such strategy may be simple and work at the item level (e.g., in multi-armed bandit it is assumed one *best fixed* arm/item exists) or implement more sophisticated RS (e.g., the objective of *A/B* testing is to find the *best fixed* RS and execute it thereafter). We argue that this assumption is rarely verified in practice, as the recommendation process itself may impact the user's preferences. For instance, a user may get bored by a strategy, while she may gain interest again, if enough time passed since the last time that strategy was used. In this case, a better approach consists in alternating different solutions at the right frequency to fully exploit their potential. In this paper, we first cast the problem as a Markov decision process, where the rewards are a linear function of the recent history of actions, and we show that a policy considering the long-term influence of the recommendations may outperform both fixed-action and contextual greedy policies. We then introduce an extension of the UCRL algorithm (LINUCRL) to effectively balance exploration and exploitation in an unknown environment, and we derive a regret bound that is independent of the number of states. Finally, we empirically validate the model assumptions and the algorithm in a number of realistic scenarios.

## 1 Introduction

Consider a movie recommendation problem, where the recommender system (RS) selects the genre to suggest to a user. A basic strategy is to estimate user's preferences and then recommend movies of the preferred genres. While this strategy is sensible in the short term, it overlooks the *dynamics* of the user's preferences caused by the recommendation process. For instance, the user may get bored of the proposed genres and then reduce her ratings. This effect is due to the recommendation strategy itself and not by an actual evolution of user's preferences, as she would still like the same genres, if only they were not proposed so often.[1]

The existence of an optimal *fixed* strategy is often assumed in RS using, e.g., matrix factorization to estimate users' ratings and the best (fixed) item/genre [16]. Similarly, multi-armed bandit (MAB) algorithms [4] effectively trade off exploration and exploitation in unknown environments, but still assume that rewards are independent from the sequence of arms selected over time and they try to select the (fixed) optimal arm as often as possible. Even when comparing more sophisticated recommendation strategies, as in *A/B* testing, we implicitly assume that once the better option (either $A$ or $B$) is found, it should be constantly executed, thus ignoring how its performance may deteriorate if used too often. An alternative approach is to estimate the *state* of the user (e.g., her

level of boredom) as a function of the movies recently watched and estimate how her preferences are affected by that. We could then learn a *contextual* strategy that recommends the best genre *depending* on the actual state of the user (e.g., using LINUCB [17]). While this could partially address the previous issue, we argue that in practice it may not be satisfactory. As the preferences depend on the sequence of recommendations, a successful strategy should "drive" the user's state in the most favorable condition to gain as much reward as possible in the long term, instead selecting the best "instantaneous" action at each step. Consider a user with preferences *1) action*, *2) drama*, *3) comedy*. After showing a few *action* and *drama* movies, the user may get bored. A greedy contextual strategy would then move to recommending *comedy*, but as soon as it estimates that *action* or *drama* are better again (i.e., their potential value reverts to its initial value as they are not watched), it would immediately switch back to them. On the other hand, a more farsighted strategy may prefer to stick to *comedy* for a little longer to increase the preference of the user for *action* to its higher level and fully exploit its potential.

In this paper, we propose to use a reinforcement learning (RL) [23] model to capture this dynamical structure, where the reward (e.g., the average rating of a genre) depends on a state that summarizes the effect of the recent recommendations on user's preferences. We introduce a novel learning algorithm that effectively trades off exploration and exploitation and we derive theoretical guarantees for it. Finally, we validate our model and algorithm in synthetic and real-data based environments.

**Related Work**. While in the MAB model, regret minimization [2] and best-arm identification algorithms [11, 22] have been often proposed to learn effective RS, they all rely on the assumption that one best fixed arm exists. [8] study settings with time-varying rewards, where each time an arm is pulled, its reward decreases due to loss of interest, but, unlike our scenario, it never increases again, even if not selected for a long time. [14] also consider rewards that continuously decrease over time whether the arm is selected or not (e.g., modeling novelty effects, where new products naturally loose interest over time). This model fits into the more general case of restless bandit [e.g., 6, 25, 20], where each arm has a partially observable internal state that evolves as a Markov chain *independently* from the arms selected over time. Time-varying preferences has also been widely studied in RS. [25, 15] consider a time-dependent bias to capture seasonality and trends effect, but do not consider the effects on users' state. More related to our model is the setting proposed by [21], who consider an MDP-based RS at the item level, where the next item reward depends on the previously $k$ selected items. Working at the item level without any underlying model assumption prevents their algorithm from learning in large state spaces, as every single combination of $k$ items should be considered (in their approach this is partially mitigated by state aggregation). Finally, they do not consider the exploration-exploitation trade-off and they directly solve an estimated MDP. This may lead to an overall linear regret, i.e., failing to learn the optimal policy. Somewhat similar, [12] propose a semi-markov model to decide what item to recommend to a user based on her latent psychological state toward this item. They assumed two possible states: sensitization, state during which she is highly engaged with the item, and boredom, state during which she is not interested in the item. Thanks to the use of a semi-markov model, the next state of the user depends on how long she has been in the current state. Our work is also related to the linear bandit model [17, 1], where rewards are a linear function of a context and an unknown target vector. Despite producing context-dependent policies, this model does not consider the influence that the actions may have on the state and thus overlook the potential of long-term reward maximization.

## 2 Problem Formulation

We consider a finite set of actions $a \in \{1, \ldots, K\} = [K]$. Depending on the application, actions may correspond to simple items or complex RS. We define the state $s_t$ at time $t$ as the history of the last $w$ actions, i.e., $s_t = (a_{t-1}, \cdots, a_{t-w})$, where for $w = 0$ the state reduces to the empty history. As described in the introduction, we expect the reward of an action $a$ to depend on how often $a$ has been recently selected (e.g., a user may get bored the more a RS is used). We introduce the recency function $\rho(s_t, a) = \sum_{\tau=1}^{w} \mathbb{1}\{a_{t-\tau} = a\}/\tau$, where the effect of an action fades as $1/\tau$, so that the recency is large if an action is often selected and it decreases as it is not selected for a while. We define the (expected) reward function associated to an action $a$ in state $s$ as

$$r(s_t, a) = \sum_{j=0}^{d} \theta_{a,j}^* \rho(s_t, a)^j = x_{s,a}^\mathsf{T} \theta_a^*, \tag{1}$$

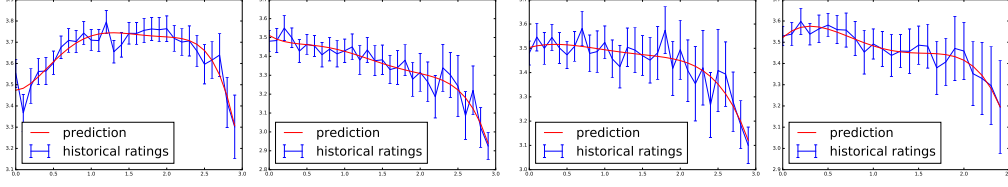

Figure 1: Average rating as a function of the recency for different genre of movies ($w = 10$) and predictions of our model for $d = 5$ in red. From left to right, *drama*, *comedy*, *action* and *thriller*. The confidence intervals are constructed based on the amount of samples available at each state $s$ and the red curves are obtained by fitting the data with the model in Eq. 1.

where $x_{s,a} = [1, \; \rho(s,a), \; \cdots, \; \rho(s,a)^d] \in \mathbb{R}^{d+1}$ is the context vector associated to action $a$ in state $s$ and $\theta_a^* \in \mathbb{R}^{d+1}$ is an unknown vector. In practice, the reward observed when selecting $a$ at $s_t$ is $r_t = r(s_t, a) + \varepsilon_t$, with $\varepsilon_t$ a zero-mean noise. For $d = 0$ or $w = 0$, this model reduces to the standard MAB setting, where $\theta_{a,0}^*$ is the expected reward of action $a$. Eq. 1 extends the MAB model by summing the "stationary" component $\theta_{a,0}^*$ to a polynomial function of the recency $\rho(s_t, a)$. While alternative and more complicated functions of $s_t$ may be used to model the reward, in the next section we show that a small degree polynomial of the recency is rich enough to model real data.

The formulation in Eq. 1 may suggest that this is an instance of a linear bandit problem, where $x_{s_t,a}$ is the context for action $a$ at time $t$ and $\theta_a^*$ is the unknown vector. Nonetheless, in linear bandit the sequence of contexts $\{x_{s,a}\}_t$ is independent from the actions selected over time and the optimal action at time $t$ is $a_t^* = \arg\max_{a \in [K]} x_{s_t,a}^{\mathsf{T}} \theta_a^*$,[2] while in our model, $x_{s_t,a}$ actually depends on the state $s_t$, that summarizes the last $w$ actions. As a result, an optimal policy should take into account its effect on the state to maximize the long-term average reward. We thus introduce the deterministic Markov decision process (MDP) $M = \langle S, [K], f, r \rangle$ with state space $S$ enumerating the possible sequences of $w$ actions, action space $[K]$, noisy reward function in Eq. 1, and a deterministic transition function $f : S \times [K] \to S$ that simply drops the action selected $w$ steps ago and appends the last action to the state. A policy $\pi : S \to [K]$ is evaluated according to its long-term average reward as $\eta^\pi = \lim_{n \to \infty} \mathbb{E}\big[1/n \sum_{t=1}^n r_t\big]$, where $r_t$ is the (random) reward of state $s_t$ and action $a_t = \pi(s_t)$. The optimal policy is thus $\pi^* = \arg\max_\pi \eta^\pi$, with optimal average reward $\eta^* = \eta^{\pi^*}$. While an explicit form for $\pi^*$ cannot be obtained in general, an optimal policy may select an action with suboptimal instantaneous reward (i.e., action $a_t = \pi(s_t)$ is s.t. $r(s_t, a_t) < \max_a r(s_t, a)$) so as to let other (potentially more rewarding) actions "recharge" and select them later on. This results into a policy that alternates actions with a fixed schedule (see Sec. 5 for more insights).[3] If the parameters $\theta_a^*$ were known, we could compute the optimal policy by using value iteration where a value function $u_0 \in \mathbb{R}^S$ is iteratively updated as

$$u_{i+1}(s) = \max_{a \in [K]} \Big[ r(s,a) + u_i\big(f(s,a)\big) \Big], \tag{2}$$

and a nearly-optimal policy is obtained after $n$ iterations as $\pi(s) = \max_{a \in [K]}[r(s,a) + u_n(f(s,a))]$. Alternatively, algorithms to compute the maximum reward cycle for deterministic MDPs could be used [see e.g., 13, 5]. The objective of a learning algorithm is to approach the performance of the optimal policy as quickly as possible. This is measured by the *regret*, which compares the reward cumulated over $T$ steps by a learning algorithm and by the optimal policy, i.e.,

$$\Delta(T) = T\eta^* - \sum_{t=1}^T r(s_t, a_t), \tag{3}$$

where $(s_t, a_t)$ is the sequence of states and actions observed and selected by the algorithm.

## 3 Model Validation on Real Data

In order to provide a preliminary validation of our model, we use the *movielens-100k* dataset [9, 7]. We consider a simple scenario where a RS directly recommends a genre to a user. In practice, one

| Genre | $d=1$ | $d=2$ | $d=3$ | $d=4$ | $d=5$ | $d=6$ |
|---|---|---|---|---|---|---|
| *action* | 0.55 | 0.74 | 0.79 | 0.81 | 0.81 | 0.82 |
| *comedy* | 0.77 | 0.85 | 0.88 | 0.90 | 0.90 | 0.91 |
| *drama* | 0.0 | 0.77 | 0.80 | 0.83 | 0.86 | 0.87 |
| *thriller* | 0.74 | 0.81 | 0.83 | 0.91 | 0.91 | 0.91 |

Table 1: $R^2$ for the different genres and values of $d$ on *movielens-100k* and a window $w = 10$.

may prefer to use collaborative filtering algorithms (e.g., matrix factorisation) and apply our proposed algorithm on top of them to find the optimal cadence to maximize long term performances. However, when dealing with very sparse information like in retargeting, it may happen that a RS just focuses on performing recommendations from a very limited set of items.[4] Once applied to this scenario, our model predicts that user's preferences change with the number of movies of the same genre a user have recently watched (e.g., she may get bored after seeing too many movies of a genre and then getting interested again as time goes by without watching that genre). In order to verify this intuition, we sort ratings for each user using their timestamps to produce an ordered sequence of ratings.[5] For different genres observed more than $10,000$ times, we compute the average rating for each value of the recency function $\rho(s_t, a)$ at the states $s_t$ encountered in the dataset. The charts of Fig. 1 provide a first qualitative support for our model. The rating for *comedy*, *action*, and *thriller* genres is a monotonically decreasing function of the recency, hinting to the existence of a boredom-effect, so that the rating of a genre decreases with how many movies of that kind have been recently watched. On the other hand, *drama* shows a more sophisticated behavior where users "discover" the genre and increase the ratings as they watch more movies, but get bored if they recently watched "too many" drama movies. This suggests that in this case there is a critical frequency at which users enjoy movies of this genre. In order to capture the dependency between rating and recency for different genres, in Eq. 1 we defined the reward as a polynomial of $\rho(s_t, a)$ with coefficients that are specific for each action $a$. In Table 1 we report the coefficient of determination $R^2$ of fitting the model of Eq. 1 to the dataset for different genres and values of $d$. The results show how our model becomes more and more accurate as we increase its complexity. We also notice that even polynomials of small degree (from $d = 4$ the $R^2$ tends to plateau) actually produce accurate reward predictions, suggesting that the recency does really capture the key elements of the state $s$ and that a relatively simple function of $\rho$ is enough to accurately predict the rating. This result also suggests that standard approaches in RS, such as matrix factorization, where the rating is contextual (as it depends on features of both users and movies/genres) but *static*, potentially ignore a critical dimension of the problem that is related to the *dynamics* of the recommendation process itself.

## 4 Linear Upper-Confidence bound for Reinforcement Learning

**The Learning Algorithm.** LINUCRL directly builds on the UCRL algorithm [10] and exploits the linear structure of the reward function and the deterministic and known transition function $f$. The core idea of LINUCRL is to construct confidence intervals on the reward function and apply the optimism-in-face-of-uncertainty principle to compute an optimistic policy. The structure of LINUCRL is illustrated in Alg. 1. Let us consider an episode $k$ starting at time $t$, LINUCRL first uses the current samples collected for each action $a$ separately to compute an estimate $\widehat{\theta}_{t,a}$ by regularized least squares, i.e.,

$$\widehat{\theta}_{t,a} = \min_{\theta} \sum_{\tau < t : a_\tau = a} \left( x_{s_\tau, a}^\mathsf{T} \theta - r_\tau \right)^2 + \lambda \|\theta\|_2, \tag{4}$$

where $x_{s_\tau, a}$ is the context vector corresponding to state $s_\tau$ and $r_\tau$ is the (noisy) reward observed at time $\tau$. Let be $R_{a,t}$ the vector of rewards obtained up to time $t$ when $a$ was executed and $X_{a,t}$ the feature matrix corresponding to the contexts observed so far, then $V_{t,a} = \left( X_{t,a}^\mathsf{T} X_{t,a} + \lambda I \right) \in \mathbb{R}^{(d+1) \times (d+1)}$ is the design matrix. The closed-form solution of the estimate is $\widehat{\theta}_{t,a} = V_{t,a}^{-1} X_{t,a}^\mathsf{T} R_{t,a}$, which gives an estimated reward function $\widehat{r}_t(s, a) = x_{s,a}^\mathsf{T} \widehat{\theta}_{t,a}$. Instead of computing the optimal

**Algorithm 1** The LINUCRL algorithm.

---

**Init:** Set $t = 0, T_a = 0, \widehat{\theta}_a = \mathbf{0} \in \mathbb{R}^{d+1}, V_a = \lambda I$
**for** rounds $k = 1, 2, \cdots$ **do**
    Set $t_k = t, \nu_a = 0$
    Compute $\widehat{\theta}_a = V_a^{-1} X_a^\mathsf{T} R_a$
    Set optimistic reward $\widetilde{r}_k(s, a) = x_{s,a}^\mathsf{T} \widehat{\theta}_a + c_{t,a} \|x_{s,a}\|_{V_a^{-1}}$
    Compute optimal policy $\widetilde{\pi}_k$ for MDP $(S, [K], f, \widetilde{r}_t)$
    **while** $\forall a \in [K], T_a < \nu_a$ **do**
        Choose action $a_t = \widetilde{\pi}_k(s_t)$
        Observe reward $r_t$ and next state $s_{t+1}$
        Update $X_{a_t} \leftarrow [X_{a_t}, x_{s_t, a_t}], R_{a_t} \leftarrow [R_{a_t}, r_t], V_{a_t} \leftarrow V_{a_t} + x_{s_t, a_t} x_{s_t, a_t}^\mathsf{T}$
        Set $\nu_{a_t} \leftarrow \nu_{a_t} + 1, t \leftarrow t + 1$
    **end while**
    Set $T_a \leftarrow T_a + \nu_a, \forall a \in [K]$
**end for**

---

policy according to the estimated reward, we compute the upper-confidence bound

$$\widetilde{r}_t(s, a) = \widehat{r}_t(s, a) + c_{t,a} \|x_{s,a}\|_{V_{t,a}^{-1}}, \tag{5}$$

where $c_{t,a}$ is a scaling factor whose explicit form is provided in Eq. 6. Since the transition function $f$ is deterministic and known, we then simply apply the value iteration scheme in Eq. 2 to the MDP $\widetilde{M}_k = \langle S, [K], f, \widetilde{r}_k \rangle$ and compute the corresponding optimal (optimistic) policy $\widetilde{\pi}_k$. It is simple to verify that $(\widetilde{M}_k, \widetilde{\pi}_k)$ is the pair of MDP and policy that maximizes the average reward over all "plausible" MDPs that are within the confidence intervals over the reward function. More formally, let $\mathcal{M}_k = \{M = \langle S, [A], f, r \rangle, \ |r(s, a) - \widehat{r}_t(s, a)| \le c_{t,a} \|x_{s,a}\|_{V_{t,a}^{-1}}, \forall s, a\}$, then with high probability we have that

$$\eta^{\widetilde{\pi}_k}(\widetilde{M}_k) \ge \max_{\pi, M \in \mathcal{M}_k} \eta^{\pi}(M).$$

Finally, LINUCRL execute $\widetilde{\pi}_k$ until the number of samples for an action is doubled w.r.t. the beginning of the episode. The specific structure of the problem makes LINUCRL more efficient than UCRL, since each iteration of Eq. 2 has $O(dSK)$ computational complexity compared to $O(S^2 K)$ of extended value iteration (used in UCRL) due to the randomness of the transitions and the optimism over $f$.

**Theoretical Analysis.** We prove that LINUCRL successfully exploits the structure of the problem to reduce its cumulative regret w.r.t. basic UCRL. We first make explicit the confidence interval in Eq. 5. Let assume that there exist (known) constants $B$ and $R$ such that $\|\theta_a^*\|_2 \le B$ for all actions $a \in [K]$ and the noise is sub-Gaussian with parameter $R$. Let $\ell_w = \log(w) + 1$, where $w$ is the length of the window in the state definition, and $L_w^2 = \frac{1 - \ell_w^{d+1}}{1 - \ell_w}$, where $d$ is the degree of the polynomial describing the reward function. Then, we run LINUCRL with the scaling factor

$$c_{t,a} = R \sqrt{(d+1) \log \left( K t^\alpha \left( 1 + \frac{T_{t,a} L_w^2}{\lambda} \right) \right)} + \lambda^{1/2} B \tag{6}$$

where $T_{t,a}$ is the number of samples collected from action $a$ up to $t$. Then we can prove the following.

**Theorem 1.** *If* LINUCRL *runs with the scaling factor in Eq. 6 over $T$ rounds, then its cumulative regret is*

$$\Delta(\text{LINUCRL}, T) \le K w \log_2 \left( \frac{8T}{K} \right) + 2 c_{\max} \sqrt{2 K T (d+1) \log \left( 1 + \frac{T L_w^2}{\lambda (d+1)} \right)},$$

*where $c_{\max} = \max_{t,a} c_{t,a}$.*

We first notice that the per-step regret $\Delta/T$ decreases to zero as $1/\sqrt{T}$, showing that as time increases, the reward approaches the optimal average reward. Furthermore, by leveraging the specific structure of our problem, LINUCRL greatly improves the dependency on other elements characterizing the MDP. In the general MDP case, UCRL suffers from a regret $O(DS\sqrt{KT})$, where $D$ is the diameter of the MDP, which in our case is equal to the history window $w$. In the regret bound of LINUCRL the

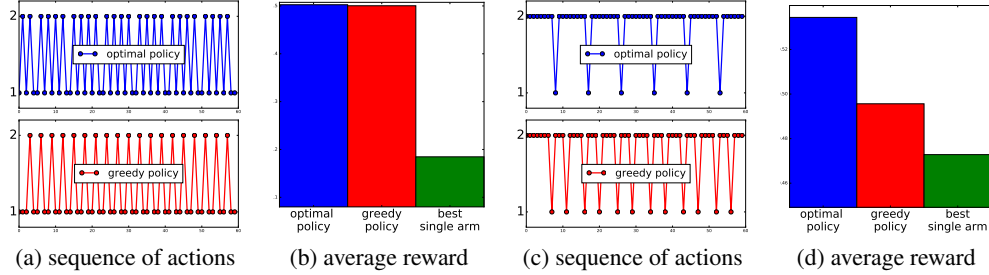

|  (a) sequence of actions | (b) average reward | (c) sequence of actions | (d) average reward |

Figure 2: Optimal policy vs. greedy and fixed-action. The fixed-action policy selects the action with the largest "constant" reward (i.e., ignoring the effects of the recommendation). The greedy policy selects the action with the highest immediate reward (depending on the state). The optimal policy is computed with value iteration. *(a-b):* parameters $c_1 = 0.3, c_2 = 0.4, \alpha = 1.5$ (limited boredom effect). *(c-d):* parameters $c_1 = 2, c_2 = 0.01, \alpha = 2$ (strong boredom effect).

dependency on the number of states (which is exponential in the history window $S = K^w$) disappears and it is replaced by the number of parameters $d + 1$ in the reward model. Furthermore, since the dynamics is deterministic and known, the only dependency on the diameter $w$ is in a lower-order logarithmic term. This result suggests that we can take a large window $w$ and a complex polynomial expression for the reward (i.e., large $d$) without compromising the overall regret. Let note that in MDPs, the worst-case regret lower bound also exhibits a $\sqrt{(T)}$ dependency ([10]), so there is not much hope to improve it. The interesting part of these bounds is actually in the problem-specific terms. Furthermore, LINUCRL compares favorably with a linear bandit approach. First, $\eta^*$ is in general much larger than the optimal average reward of a greedy policy selecting the best instantanous action at each step. Second, apart from the $\log(T)$ term, the regret is the same of a linear bandit algorithm (e.g., LINUCB). This means that LINUCRL approaches a better target performance $\eta^*$ almost at the same speed as linear bandit algorithms reach a worse greedy policy. Finally, [19] developed a specific instance of UCRL for deterministic MDPs, whose final regret is of order $O(\lambda A \log(T)/\Delta)$, where $\lambda$ is the length of the largest simple cycle that can be generated in the MDP and $\Delta$ is the gap between the reward of the optimal and second-optimal policy. While the regret in this bound only scales as $O(\log T)$, in our setting $\lambda$ can be as large as $S = K^w$, which is exponentially worse than the diameter $w$, and $\Delta$ can be arbitrarily small, thus making a $O(\sqrt{T})$ bound often preferable. We leave the integration of our linear reward assumption into the algorithm proposed by [19] as future work.

## 5 Experiments

In order to validate our model on real datasets, we need persistent information about a user identification number to follow the user through time and evaluate how preferences evolve over time in response to the recommendations. This also requires datasets where several RSs are used for the same user with different cadence and for which it is possible to associate a user-item feedback with the system that actually performed that recommendation. Unfortunately, these requirements make most of publicly available datasets not suitable for this validation. As a result, we propose to use both synthetic and dataset-based experiments to empirically validate our model and compare LINUCRL to existing baselines. We consider three different scenarios. *Toy experiment:* A simulated environment with two actions and different parameters, with the objective of illustrating when the optimal policy could outperform fixed-action and greedy strategies. *Movielens:* We derive model parameters from the *movielens* dataset and we compare the learning performance (i.e., cumulative reward) of LINUCRL to baseline algorithms. *Real-world data from A/B testing:* this dataset provides enough information to test our algorithm and although our model assumptions are no longer satisfied, we can still investigate how a long-term policy alternating $A$ and $B$ on the basis of past choices can outperform each solution individually.

**Optimal vs. fixed-action and greedy policy**. We first illustrate the potential improvement coming from a non-static policy that takes into consideration the recent sequence of actions and maximizes the long-term reward, compared to a greedy policy that selects the action with the higher immediate reward at each step. Intuitively, the gap may be large whenever an action has a large instantaneous reward that decreases very fast as it is selected (e.g., boredom effect). A long-term strategy may prefer to stick to selecting a sub-optimal action for a while, until the better action goes back to its

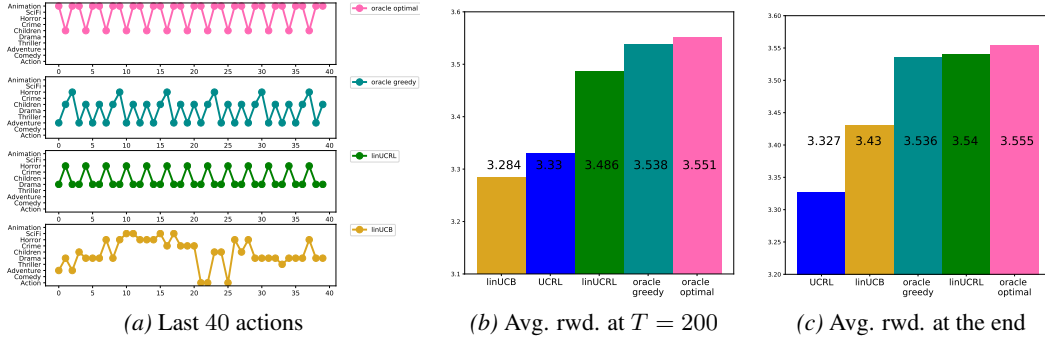

*(a)* Last 40 actions      *(b)* Avg. rwd. at $T = 200$      *(c)* Avg. rwd. at the end

Figure 3: Results of learning experiment based on *movielens* dataset.

initial value. We consider the simple case $K = 2$ and $d = 1$. Let $\theta_1^* = (1, c_1), \theta_2^* = (1/\alpha, c_2)$. We study the optimal policy maximizing the average reward $\eta$, a greedy policy that always selects $a_t = \arg\max_a r(s_t, a)$, and a fixed-action policy $a_t = \arg\max\{1, 1/\alpha\}$. We first set $c_1 = 0.3 \approx c_2 = 0.4$ and $\alpha = 1.5$, for which the "boredom" effect (i.e., the decrease in reward) is very mild. In this case (see Fig. 2-*(left)*), the fixed-action policy performs very poorly, while greedy and optimal policy smartly alternates between actions so as to avoid decreasing the reward of the "best" action too much. In this case, the difference between greedy and optimal policy is very narrow. However in Fig. 2-*(right)*, with $c_1 = 2 \gg c_2 = 0.01$ and $\alpha = 2$, we see that the greedy policy switches to action 1 too soon to gain immediate reward (plays action 1 for $66\%$ of the time) whereas the optimal policy stick to action 2 longer (plays action 1 for $57\%$ of the time) so as to allow action 1 to regain reward and then go back to select it again. As a result, the optimal policy exploits the full potential of action 1 better and eventually gains higher average reward. While here we only illustrate the "boredom" effect (i.e., the reward linearly decreases with the recency), we can imagine a large range of scenarios where the greedy policy is highly suboptimal compared to the optimal policy.

**Learning on *movielens* dataset.** In order to overcome the difficulty of creating full complex RS and evaluate them on offline datasets, we focus on a relatively simple scenario where a RS directly recommends movies from one chosen genre, for which we have already validated our model in Sec. 3. One strategy could be to apply a bandit algorithm to find the optimal genre and then always recommend movies of this genre. On the other hand, our algorithm tries to identify an optimal sequence of those genres to keep the user interested. The standard offline evaluation of a learning algorithm on historical data is to use a replay or counterfactual strategy [18, 24], which consists in updating the model whenever the learning algorithm takes the same action as in the logged data, and only update the state (but not the model) otherwise. In our case this replay strategy cannot be applied because the reward depends on the history of selected actions and we could not evaluate the reward of an action if the algorithm generated a sequence that is not available in the dataset (which is quite likely). Thus in order to compare the learning performance of LINUCRL to existing baselines, we use the *movielens100k* dataset to estimate the parameters of our model and construct the corresponding "simulator". Unlike a fully synthetic experiment, this gives a configuration which is "likely" to appear in practice, as the parameters are directly estimated from real data. We choose $K = 10$ actions corresponding to different genres of movies, and we set $d = 5$ and $w = 5$, which results into $K^w = 10^5$ states. We recall that $w$ has a mild impact on the learning performance of LINUCRL as it does not need to repeatedly try the same action in each state (as UCRL) to be able to estimate its reward. This is also confirmed by the regret analysis that shows that the regret only depends on $w$ in the lower-order logarithmic term of the regret. Given this number of states, UCRL would need at least one million iteration to observe each state 10 times which is dramatically too large for the application we consider. The parameters that describe the dependency of the reward function on the recency (i.e., $\theta_{j,a}^*$) are computed by using the ratings averaged over all users for each state encountered and for ten different genres in the dataset. The first component of the vectors $\theta_a^*$ is chosen to simulate different user's preferences and to create complex dynamics in the reward functions. The resulting parameters and reward functions are reported in App. B. Finally, the observed reward is obtained by adding a small random Gaussian noise to the linear function. In this setting, a constant strategy would always pull the comedy genre since it is the one with the highest "static" reward, while other genres are also highly rewarding and a suitable alternation between them may provide a much higher reward.

We compare LINUCRL to the following algorithms: *oracle optimal* ($\pi^*$), *oracle greedy* (greedy contextual policy), LINUCB [1] (learn the parameters using LINUCB for each action and select the

| Algorithm | on the $T$ steps | on the last steps |
|---|---|---|
| only B | 46.0% | 46.0% |
| UCRL | 46.5% | 46.0% |
| LINUCRL | 66.7% | 75.8% |
| oracle greedy | 61.3% | 61.3% |
| oracle optimal | 95.2% | 95.2% |

Table 2: Relative improvement over *only A* of learning experiment based on *large scale A/B testing* dataset.

one with largest instantaneous reward), UCRL [3] (considering each action and state independently). The results are obtained by averaging 4 independent runs. Fig. 3*(*b-c) shows the average reward at $T = 200$ and after $T = 2000$ steps. We first notice that as in the previous experiment the oracle greedy policy is suboptimal compared to the optimal policy that maximizes the long-term reward. Despite the fact that UCRL targets this better performance, the learning process is very slow as the number of states is too large. Indeed this number of steps is lower than the number of states so UCRL did not have the chance to update its policy since in average no states has been visited twice. On the other hand, at early learning stages LINUCRL is already better than LINUCB, and its performance keeps improving until, at 2000 steps, it actually performs better than the oracle greedy strategy and it is close to the optimal policy.

**Large scale *A/B* testing dataset.** We also validate our approach on a real-world *A/B* testing dataset. We collected 15 days of click on ads history of a CRITEO's test, where users have been proposed two variations on the display denoted as $A$ and $B$. Each display is actually the output of two real-world collaborative-filtering recommender strategies; precise information on how these algorithms are constructed is not relevant for our analysis. Unlike a classical *A/B* testing each unique user has been exposed to *both* $A$ and $B$ but with different frequencies. This dataset is formed of 350M tuples *(user id, timestamp, version, click)* and will be released publicly as soon as possible. Remark that the system is already heavily optimized and that even a small improvement in the click-rate is very desirable. As in the *movielens* experiment, we do not have enough data to evaluate a learning algorithm on the historical events (not enough samples per state would be available), so we first compute a simulator based on the data and then run LINUCRL- that does not know the parameters of the simulator and must try to estimate them - and compare it to simple baselines. Unlike the previous experiment, we do not impose any linear assumption on the simulator (as in Eq. 1) and we compute the click probability for actions $A$ and $B$ independently in each state (we set $w = 10$, for a total of $2^{10} = 1024$ states) and whenever that state-action pair is executed we draw a Bernoulli with the corresponding probability. Using this simulator we compute oracle greedy and optimal policies and we compare LINUCB, LINUCRL, which is no longer able to learn the "true" model, since it does not satisfy the linear assumption, and UCRL, which may suffer from the large number of state but targets a model with potentially better performance (as it can correctly estimate the actual reward function and not just a linear approximation of it). We report the results (averaged over 5 runs) as a relative improvement over the worst fixed option (i.e., in this case $A$). Tab. 2 shows the average reward over $T = 2,000$ steps and of the learned policy at the end of the experiment. Despite the fact that the simulator does not satisfy our modeling assumptions, LINUCRL is still the most competitive algorithm as it achieves the best performance among the learning algorithms and it outperforms the oracle greedy policy.

## 6 Conclusion

We showed that estimating the influence of the recommendation strategy on the reward and computing a policy maximizing the long-term reward may significantly outperform fixed-action or greedy contextual policies. We introduced a novel learning algorithm, LINUCRL, to effectively learn such policy and we prove that its regret is much smaller than for standard reinforcement learning algorithms (UCRL). We validated our model and its usefulness on the *movielens* dataset and on a novel *A/B* testing dataset. Our results illustrate how the optimal policy effectively alternates between different options, in order to keep the interest of the users as high as possible. Furthermore, we compared LINUCRL to a series of learning baselines on simulators satisfying our linearity assumptions (*movielens*) or not (*A/B* testing). A venue for future work is to extend the current model to take into consideration correlations between actions. Furthermore, given its speed of convergence, it could be interesting to run a different instance of LINUCRL per user - or group of users - in order to offer personalized "boredom" curves. Finally, using different models of the reward as a function of the recency (e.g., logistic regression) could be used in case of binary rewards.

## Footnotes

[1]In this paper, we do not study non-stationarity preferences, as it is a somehow orthogonal problem to the issue that we consider.

[2]We will refer to this strategy as "greedy" policy thereafter.

[3]In deterministic MDPs the optimal policy is a recurrent sequence of actions inducing a maximum-reward cycle over states.

[4]See Sect. 5 for further discussion on the difficulty of finding suitable datasets for the validation of time-varying models.

[5]In the movielens dataset a timestamp does not correspond to the moment the user saw the movie but when the rating is actually submitted. Yet, this does not cancel potential dependencies of future rewards on past actions.

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

# A Proof of Theorem 1

**Proof.** In order to prove Thm. 1, we first need the following proposition about the confidence intervals used in computing the optimistic reward $\widetilde{r}(s,a)$.

**Proposition 2.** *Let assume* $\|\theta_a^*\|_2 \leq B$. *If* $\widehat{\theta}_{t,a}$ *is computed as in Eq. 4 and* $c_{t,a}$ *is defined as in Eq. 6, then*

$$\mathbb{P}\left(r(s,a) \leq \widehat{r}(s,a) + c_{t,a}\|x_{s,a}\|_{V_{t,a}^{-1}}\right) \leq \frac{t^{-\alpha}}{K}.$$

*Proof.* By definition of $\rho(s,a)$ we have $0 \leq \rho(s,a) \leq \sum_{\tau=1}^{w}\frac{1}{\tau} < \log(w) + 1 \doteq \ell_w$. Thus $1 \leq \|x_{s,a}\|_2^2 \leq \sum_{j=0}^{d}\ell_w^j = \frac{1-\ell_w^{d+1}}{1-\ell_w} = L_w^2$. Using Thm. 2 of [1], we have with probability $1 - \delta$,

$$\|\widehat{\theta}_{t,a} - \theta_a^*\|_{V_{t,a}} \leq R\sqrt{(d+1)\log\left(\frac{1+T_{t,a}L_w^2/\lambda}{\delta}\right)} + \lambda^{1/2}B.$$

Thus for all $s \in S$ we have,

$$|r(s,a) - \widehat{r}(s,a)| = |x_{s,a}^\mathsf{T}\widehat{\theta}_{t,a} - x_{s,a}^\mathsf{T}\theta_a^*| \leq \|x_{s,a}\|_{V_{t,a}^{-1}}\|\widehat{\theta}_a - \theta_a^*\|_{V_{t,a}}.$$

Using $\delta = \frac{t^{-\alpha}}{K}$ concludes the proof. $\square$

An immediate result of Prop. 2 is that the estimated average reward of $\widetilde{\pi}_k$ in the optimistic MDP $\widetilde{M}_k$ is an upper-confidence bound on the optimal average reward, i.e., for any $t$ (the probability follows by a union bound over actions)

$$\mathbb{P}\left(\eta^* > \eta^{\widetilde{\pi}_k}(\widetilde{M}_k)\right) \leq t^{-\alpha}. \tag{7}$$

We are now ready to prove the main result.

*Proof of Thm. 1.* We follow similar steps as in [10]. We split the regret over episodes as

$$\Delta(\mathcal{A}, T) = \sum_{k=1}^{m}\sum_{t=t_k}^{t_{k+1}-1}\left(\eta^* - r(s_t, a_t)\right) = \sum_{k=1}^{m}\Delta_k.$$

Let $\mathcal{T}_{k,a} = \{t_k \leq t < t_{k+1} : a_t = a\}$ be the steps when action $a$ is selected during episode $k$. We upper bound the per-episode regret as

$$\Delta_k = \sum_{a\in[K]}\sum_{t\in\mathcal{T}_{k,a}}\left(\eta^* - r(s_t, a)\right) \leq \sum_{t=t_k}^{t_{k+1}-1}\left(\widetilde{\eta}_k - \widetilde{r}_k(s_t, a)\right) + \sum_{a\in[K]}\sum_{t\in\mathcal{T}_{k,a}}\left(\widetilde{r}_k(s_t, a) - r(s_t, a)\right),$$

where the inequality directly follows from the event that $\widetilde{\eta}_k \geq \eta^*$ (Eq. 7) with probability $1 - T^{-\alpha}$. Notice that the low-probability event of failing confidence intervals can be treated as in [10].

We proceed by bounding the first term of Eq. 8. Unlike in the general online learning scenario, in our setting the transition function $f$ is known and thus the regret incurred from bad estimates of the dynamics is reduced to zero. Furthermore, since we are dealing with deterministic MDPs, the optimal policy converges to a loop over states. When starting a new policy, we may start from a state outside its loop. Nonetheless, it is easy to verify that starting from any state $s$, it is always possible to reach any desired state $s'$ in at most $w$ steps (i.e., the size of the history window). As a result, within each episode $k$ the difference between the cumulative reward ($\sum_t \widetilde{r}_k(s_t, a)$) and the (optimistic) average reward (($t_{k+1} - t_k)\widetilde{\eta}_k$) in the loop never exceeds $w$. Furthermore, since episodes terminate when one action doubles its number of samples, using a similar proof as [10], we have that the number of episodes is bounded as $m \leq K\log_2(\frac{8T}{K})$. As a result, the contribution of the first term of Eq. 8 to the overall regret is bounded as

$$\sum_{k=1}^{m}\sum_{t=t_k}^{t_{k+1}-1}\left(\widetilde{\eta}_k - \widetilde{r}_k(s_t, a)\right) \leq Kw\log_2\left(\frac{8T}{K}\right). \tag{8}$$

The second term in Eq. 8 refers to the (cumulative) reward estimation error and it can be decomposed as

$$|\widetilde{r}_k(s_t, a) - r(s_t, a)| \leq |\widetilde{r}_k(s_t, a) - \widehat{r}_k(s_t, a)| + |\widehat{r}_k(s_t, a) - r(s_t, a)|.$$

We can bound the cumulative sum of the second term as (similar for the first, since $\widetilde{r}_k$ belongs to the confidence interval of $\widehat{r}_k$ by construction)

$$\sum_{k=1}^{m} \sum_{a \in [K]} \sum_{t \in \mathcal{T}_{k,a}} |\widehat{r}_k(s_t, a) - r(s_t, a)| \leq \sum_{k=1}^{m} \sum_{a \in [K]} \sum_{t \in \mathcal{T}_{k,a}} c_{t,a} \|x_{s_t,a}\|_{V_{a,t}^{-1}}$$

$$\leq c_{\max} \sum_{a \in [K]} \sqrt{\sum_{k=1}^{m} \sum_{t \in \mathcal{T}_{k,a}} \|x_{s_t,a}\|_{V_{a,t}^{-1}}^2} \sqrt{T_a},$$

where the first inequality follows from Prop. 2 with probability $1 - T^{-\alpha}$, and $T_a$ is the total number of times $a$ has been selected at step $T$. Let $\mathcal{T}_a = \cup_k \mathcal{T}_{k,a}$, then using Lemma 11 of [1], we have

$$\sum_{t \in \mathcal{T}_a} \|x_{s_t,a}\|_{V_{t,a}^{-1}}^2 \leq 2 \log \frac{\det(V_{T,a})}{\det(\lambda I)},$$

and by Lem. 10 of [1], we have

$$\det(V_{t,a}) \leq (\lambda + t L_w^2/(d+1))^{d+1},$$

which leads to

$$\sum_{k=1}^{m} \sum_{a \in [K]} \sum_{t \in \mathcal{T}_{k,a}} |\widehat{r}_k(s_t, a) - r(s_t, a)| \leq c_{\max} \sum_{a \in [K]} \sqrt{T_a} \sqrt{2(d+1) \log \left( \frac{\lambda + t L_w^2}{\lambda(d+1)} \right)}$$

$$\leq c_{\max} \sqrt{2KT(d+1) \log \left( \frac{\lambda + t L_w^2}{\lambda(d+1)} \right)}.$$

Bringing all the terms together gives the regret bound. □

# B  Experiments Details

| Genre | $\theta_{a,0}^*$ | $\theta_{a,1}^*$ | $\theta_{a,2}^*$ | $\theta_{a,3}^*$ | $\theta_{a,4}^*$ | $\theta_{a,5}^*$ |
|---|---|---|---|---|---|---|
| *Action* | 3.1 | 0.54 | -1.08 | 0.78 | -0.22 | 0.02 |
| *Comedy* | 3.34 | 0.54 | -1.08 | 0.78 | -0.22 | 0.02 |
| *Adventure* | 3.51 | 0.86 | -2.7 | 3.06 | -1.46 | 0.24 |
| *Thriller* | 3.4 | 1.26 | -2.9 | 2.76 | -1.14 | 0.16 |
| *Drama* | 2.75 | 1.0 | 0.94 | -1.86 | 0.94 | -0.16 |
| *Children* | 3.52 | 0.1 | 0.0 | -0.3 | 0.2 | -0.04 |
| *Crime* | 3.37 | 0.32 | 1.12 | -3.0 | 2.26 | -0.54 |
| *Horror* | 3.54 | -0.68 | 1.84 | -2.04 | 0.82 | -0.12 |
| *SciFi* | 3.3 | 0.64 | -1.32 | 1.1 | -0.38 | 0.02 |
| *Animation* | 3.4 | 1.38 | -3.44 | 3.62 | -1.62 | 0.24 |

Table 3: Reward parameters of each genre for the *movielens* experiment.

The parameters used in the MovieLens experiment are reported in Table 3.

