[Reviews · NeurIPS 2018]

Reviewer 1



The paper proposes an algorithm, LINUCRL that recognizes the fact that users' preferences can be time-varying and the recommendations may itself influence the preferences. I have the following questions/ suggestions a) The paper mainly cites several key papers in MAB setting. However, boredom in recommendation has been previously studied. For instance, WSDM 2015 paper "Just in Time Recommendations: Modeling the Dynamics of Boredom in Activity Streams". Citing some of the work addressing boredom and contrasting this paper's contribution could be useful. b) The authors illustrate the farsightedness with the three genre example of action, drama and comedy. Churn is a serious problem for online recommendation (a user may get frustrated with the recommendations and leave the system). How does the authors plan to address churn? c) I like the way the paper attempts to circumvent unavailability of data sets and construct the experiments. However, I felt the algorithmic contribution of the paper is not substantial. Also, I would like the recommendations to be more fine-grained than just plain genre and extending the results to more fine-grained recommendations would improve the paper. Overall, I find the paper well-written, while the literature review section can certainly expand few lines of past work involving boredom. %After rebuttal% After reading the authors' rebuttal, I am willing to increase the overall score to 6.

Reviewer 2



The paper changes the model assumptions for recommender systems (RS) to capture phenomena of real world data that has been largely ignored up to now. Instead of assuming a fixed preference over time, the utility of a recommendation is based on the frequency of previous occurrences in a time window w. This captures the fact that humans get bored of repetition. The authors do a great job in showing that this effect occurs in real data. However they still make quite restrictive model assumptions, EDIT{misunderstood this part in the paper, remove comment: i.e. that the utility of an action is only based on the frequency it occurred over the last w times, without taking the positioning into account. The question is whether their model is a good trade-off between computational feasibility and how well it captures the data. To complete the picture, it would be nice to see how much gain in accuracy there is from considering more complex functions that depend on when exactly a recommendation was made. } The main contribution, beside defining a novel model, is an adaptation of UCRL to this problem. The algorithm exploits that the setting is reduced to a deterministic MDP with linear reward function. They use standard techniques to proof an upper bound for this problem, that has favourable dependencies on the problem parameters. The math seems sound, but I haven't checked every detail in the appendix. The paper is well written and easy to understand. The performance of the algorithm is verified in several experiments, which are semi-synthetic. The experimental setup is reasonable and shows convincingly the advantages of the new model. The biggest downside of this algorithm is, that it only works for small a window w. The authors correctly note that the regret scales fine in w. But they should highlight that the run-time is actually exponential in w. Thereby making this algorithm computationally infeasible even for moderate window sizes.

Reviewer 3



This work proposes LINUCRL which extends previous work UCRL to dynamically capture users’ preference in recommender systems. The experiments on synthetic and real-world dataset show the proposed method can effectively trade-off exploration and exploitation. The theoretical analysis of LINUNCRL is provided. (1) While Table 2 shows that LINUCRL achieves better performance as compared to baselines, some strong RL-based recommendation baselines are missing. (2) Is this approach also work for the cold-start recommendation, i.e., when the history of action becomes extremely sparse or completely not available? (3) The advantages of the proposed method are not very clear as compared to existing methods. I read the response, and some of my concerns are addressed. I keep my score unchanged but I think some RL-based recommendation models could be also considered, e.g., [1] Time-Sensitive Recommendation From Recurrent User Activities [2] Novelty Learning via Collaborative Proximity Filtering